# An Efficient Extended Targets Detection Framework Based on Sampling and Spatio-Temporal Detection

**DOI:** 10.3390/s19132912

**Published:** 2019-07-01

**Authors:** Bo Yan, Na Xu, Wenbo Zhao, Muqing Li, Luping Xu

**Affiliations:** 1School of Aerospace Science and Technology, XIDIAN University, 266 Xinglong Section of Xifeng Road, Xi’an 710126, China; 2School of Life Sciences and Technology, XIDIAN University, 266 Xinglong Section of Xifeng Road, Xi’an 710126, China

**Keywords:** marine radar system, target detection, extended target, clutter suppression

## Abstract

Excellent performance, real-time and low memory requirement are three vital requirements for target detection in high resolution marine radar system. Unfortunately, many current state-of-the-art methods merely achieve excellent performance when coping with highly complex scenes. In fact, a common problem is that real-time processing, low memory requirement and remarkable detection ability are difficult to coordinate. To address this issue, we propose a novel detection framework which bases its principle on sampling and spatiotemporal detection. The framework consists of two stages, coarse detection and fine detection. Sampling-based coarse detection is designed to guarantee the real-time processing and low memory requirements by locating the area where targets may exist in advance. Different from former detection methods, multi-scan video data are utilized. In the stage of fine detection, the candidate areas are grouped into three categories: single target, dense targets and sea clutter. Different approaches for processing the different categories are implemented to achieve excellent performance. The superiority of the proposed framework beyond state-of-the-art baselines is well substantiated in this work. Low memory requirement of the proposed framework was verified by theoretical analysis. Real-time processing capability was verified by the video data of two real scenarios. Synthetic data were tested to show the improvement in tracking performance by using the proposed detection framework.

## 1. Introduction

Moving target detection plays a primary and pivotal role in the marine radar system, which aims to completely and accurately detect moving objects from video data. For the non-Gaussian sea clutter and complex backgrounds, using sequential radar images to extract targets of interest such as vessels and low-flying aircraft is a challenging task. The moving target detection problem mainly has two issues, target detection and target tracking. Target detection is explored to find the candidate positions of targets by the video data originated from radar front-end. Target tracking is designed to associate the positions into the trajectories of the targets. As the increased resolution of modern radar, targets would be found in several resolution cells rather than merely appearing in one single resolution cell. Then, the high-resolution radar would receive more than one points per time step from different corner reflectors of a single target. The target is unsuitable to be categorized as a point. Therefore, a hot research topic, extended target detection and tracking, arises recently. The aim of this work is to develop a novel target detection framework to improve the extended target tracking performance by providing more accurate points of targets and fewer false alarm points. 

Various algorithms have been developed for multiple extended target tracking (METT). ET-PHD-based algorithms [1,2,3,4,5] are capable of estimating the target extent and measurement rates as well as the kinematic state of the target. For the weak extended target, track-before-detect (TBD) methodology, which makes full use of multi-scan, has been employed to develop Hough transformation-based TBD [6], dynamic programming-based TBD [7], Grey Wolf optimization-based (GWO) TBD [8], and particle filter-based methods [9]. The existing methods [1,2,3,4,5,6,7,8,9] have achieved excellent tracking performance. It is hard to further significantly improve tracking performance by developing delicate tracking algorithms. Therefore, providing more accurate points of targets and less false alarm points by improved extended target detection method is promising to improve the performance of radar systems. The PHD-based filters [1,2,3,4,5] are fed with the points provided by an originally ordered statistic constant false alarm rate (OS-CFAR) detector [10]. Following the work in [10], more improved versions of CFAR detectors [11,12,13] are developed. Wang C. et al. [11] present an intensity-space (IS) domain CFAR ship detector. In [12], clutter modeling has been identified as a viable solution to suppress false detection. Gao G. et al. [13] develope a statistical model of the filter in nonhomogeneous sea clutter to achieve CFAR detection. However, due to the presence of non-intentional interference (sea clutter, thermal clutter and ground clutter) and the echoes of background (mountains, shores, buildings, islands and motor vehicles), many false alarms exist. The methods in [10,11,12,13] are insufficient in suppressing the fixed clutter. To address this problem, clutter map-based CFAR (CM-CFAR) methods [14,15], which take the full benefits of multi-scans, are proposed. Conte et al. [14] develop a CM-CFAR relying on a combination of space and time processing. In [15], the background noise/clutter power is estimated by processing the returns in the map cell from all scans up to the current one. Using temporal information and spatial information simultaneously is another powerful technology that benefits from multi-scans for clutter suppression [16,17,18]. The priority of the methods is identifying the background priors for a video [16]. The spatial saliency map and the temporal saliency map are calculated in [17] and spatiotemporal local contrast filter is developed in [18]. However, drawbacks still exist. Pixels of radar video increase dramatically for the increases of resolution and coverage range. Many more calculations and memory are required for using the detection methods [10,11,12,13,14,15,16,17,18]. Meanwhile, one frame of video must be processed within a radar scanning cycle by limited memory space. However, the methods in [10,11,12,13,14,15,16,17,18] cannot process the video in real time. Meanwhile, the memory requirements of the methods in [14,15,16,17,18] are enormous. The above-mentioned shortcomings drastically limit the utilization of the methods in [10,11,12,13,14,15,16,17,18] in engineering. 

A series of algorithms has been developed in our former works [19,20,21,22] to address excellent performance, real-time processing, and low memory requirement simultaneously. A contour tracking algorithm is used in [19] to meet the real-time processing in advance. The detection approach in [20], which uses a region growing algorithm, is developed to improve location precision. The methods in [21,22] are designed to detect targets in dense targets scenarios such as fleet detection and targets in lanes. To suppress the fixed clutter, an efficient spatiotemporal detection method based on sampling is designed. Meanwhile, the sampling-based spatiotemporal detection method and the methods in [20,21,22] are integrated into the novel detection framework. The methods in [19,20,21] detect targets only by the current frame. In this work, both the current frame and several past frames are utilized to estimate the intensity of the clutter. Compared with former work [19,20,21], more video data are utilized to improve the detection performance. Thus, our former works [19,20,21,22] are components in the proposed framework. We do not simply piece together these components. The framework is designed to make each component work well with the others. Past frames have not been used in extended target detection before and the detection methods are not combined with target tracking methods in [19,20,21,22]. Their excellent performance can be improved by fine detection. Meanwhile, computation and memory requirements can be decreased by coarse detection. In the first stage, a sampled map evaluating the clutter intensity of surveillance area is built to suppress the fixed clutter. Unlike spatiotemporal-based filters [18], little memory is required for sampling on range, azimuth and time axes. The coarse detection is designed to roughly locate the area where targets may exist in advance by uniformly selecting seeds in the whole surveillance area. Only the selected seeds are used to guarantee the real-time processing and low memory requirement. In the fine detection stage, only the areas where targets may exist are processed. The candidate areas are identified into three categories, namely single target, dense targets and sea clutter, by the contours of the areas [21]. The areas of dense targets are further separated into subareas using the Rain algorithm method [22]. Each subarea is regarded as an individual target. Excellent performance can be achieved by the fine detection. As presented in Figure 1, the input of the target detection is the video sequences of radar. The results of target detection are three-dimensional points, i.e. two-dimensional positional information and its measuring time. The measuring time in target tracking algorithms can be simply represented by the frame number (see, e.g., [1,2,3,4]). Correct points can be obtained by the detection framework to further improve the final tracking performance. Figure 1 describes the relationship between the radar data processing and the existing methods mentioned above.

The remainder of the work is organized as follows. Section 2 defines the models and problems. Section 3 presents the implementation of the sampling-based spatiotemporal detection method. In this section, the proposed detection framework is also presented. The superiority of the proposed framework beyond state-of-the-art baselines is substantiated in Section 4 using real high-resolution marine radar data as well as synthetic data. Section 5 draws conclusions.

## 2. Models and Notations

### 2.1. Target Model

Assume that the extended targets are randomly distributed on an x–y plane. We use M_k_ to denote the number of targets at *k*^th^ scan. The size and quantity of targets are unknown. A general approach based on support functions to model smooth object shapes presented in [23] is used here. The state of an individual extended target can be modeled in state space ***R****^s^*, *s* = 6. The target state of *m*^th^ target is *St_m_* = (*x_m_*, *y_m_*, *l_m_^’^*, *w_m_^’^*, *α_m_*, *p_m_*), 1 ≤ *m* ≤ *M*. *x_m_* and *y_m_* denote the centroid of *m*^th^ target on x- and y-axis, respectively. *l_m_*^’^ and *w_m_*^’^ are the lengths of the major axis and the minor axis. *α_m_* is the angle between the major axis and line of sight. *p_m_* denotes the intensity present in a single pulse return. The comparison between reflection models and real data [20] infers that Swerling type 1 is more appropriate to express the magnitude of the target. The magnitude of a pulse *y_t_* return follows the Rayleigh distribution.
(1)$$f_{t}(y_{t})=\frac{2y_{t}}{p}\exp(-\frac{y_{t}{}^{2}}{p}),y_{t}>0$$
where *y_t_* means the intensity of the echo originated from a target. The intensity of a target in an azimuth bin *M*(*r,a*) is presented in Figure 2.

### 2.2. Noise Model

The clutter consists of two parts, sea clutter and measurement noise. The mean and variance of sea clutter are closely related to the sea state. The sea clutter distribution model of major theoretic and practical interest is the so-called *K*-distributed clutter model [24,25,26], and the PDF (probability distribution function) is Equation (2).
(2)$$f_{c}(p_{c})=\frac{2b}{\Gamma (v+1)}{\left(\frac{bp_{c}}{2}\right)}^{v+1}K_{v}(bp_{c}),$$
where *p_c_* denotes the power of the sea clutter in this cell, Γ(*v*) is the gamma function, ν is referred to as the shape parameter, *b* is a scale parameter, and *K_v_*(*u*) denotes the modified Bessel function of second kind and order. Measurement noise is a zero-mean, white and uncorrelated Gaussian noise sequence.

### 2.3. Measurement Model 

The surveillance area is divided into *N_A_* × *N_R_* grid cells in a polar coordinate, where *N_A_* and *N_R_* are the number of cells on azimuth and range axes, respectively. Each cell corresponds to a pixel in radar video. Figure 2 show that the video data can be modeled by Equation (3) in a polar coordinate. Parameters *r* and *a* denote the location on range and azimuth axes. *Z*(*r,a*) means the amplitude in the range–azimuth resolution cell (*r,a*).
(3)$$Z(r,a)=N(r,a)+\int_{-\pi }^{\pi }\omega (\theta )(C(r,a+d\theta )+M(r,a+d\theta ))d\theta ,$$
where *ω*(*θ*) in Figure 2 is the antenna pattern function. The non-noise measurement of cell *Z*(*r, a*) is related to the RCS of target *M*(*r, a*) and the clutter *C*(*r, a*). The distribution of *M*(*r, a*) is related to the shape and material of the target in cell (*r, a*). *N*(*r, a*) denotes the additive measurement noise. 

In Figure 2, an aircraft is illuminated by radar beams. The measurement model of marine radar [20,21] infers that, once parts of the aircraft are illuminated by the beam, the radar echoes in this azimuth bin are affected by the aircraft. The target would be illuminated by the main lobe of the beam when the direction of the beam equals *φ*. The scope of *φ*, *φ*_1_ ≤ *φ* ≤ *φ*_2_, can be estimated by Equation (4). The area where the echo is affected by the extended target in azimuth and range axes can be represented by *A* and *R,* respectively.
(4)$$\begin{array}{ll} A=\varphi_{2}-\varphi_{1} & =2\psi +\theta_{0}\\ & \approx \;2\arccos(\frac{{\left(x_{k}{}^{\prime }\right)}^{2}+{\left(y_{k}{}^{\prime }\right)}^{2}-x_{k}{}^{\prime }l^{\prime }\cos\alpha_{k}-y_{k}{}^{\prime }l^{\prime }\sin\alpha_{k}}{\sqrt{{\left(x_{k}{}^{\prime }\right)}^{2}+{\left(y_{k}{}^{\prime }\right)}^{2}}\sqrt{{\left(x_{k}{}^{\prime }-l^{\prime }\cos\alpha_{k}\right)}^{2}+{\left(y_{k}{}^{\prime }-l^{\prime }\sin\alpha_{k}\right)}^{2}}})+\theta_{0} \end{array}$$

The proof of Equation (4) is presented in the Appendix of [7]. *θ_0_* denotes 3dB azimuth beam width of the radar. The expression of *A* and *R* is presented by
(5)$$\{\begin{array}{l} \theta_{0}\leq A\leq 2\arctan(\frac{l_{\max}{}^{\prime }}{\sqrt{{\left(x_{k}{}^{\prime }\right)}^{2}+{\left(y_{k}{}^{\prime }\right)}^{2}}})+\theta_{0}\\ 0\leq R\leq l_{\max}{}^{\prime } \end{array}$$

*l_max_* and *l_min_* are the upper and lower limits of *l^’^*. Equation (4) infers that, for the extension of the beams, the image of the target is larger than its real size. The azimuth bins and range bins whose amplitude (*a_m_*, *r_m_*) might be affected by the object can be estimated using Equation (6).
(6)$$\{\begin{array}{l} \left\lceil \frac{(2\arctan(\frac{l_{\min}}{\sqrt{{\left(x_{k}{}^{\prime }\right)}^{2}+{\left(y_{k}{}^{\prime }\right)}^{2}}})+\theta_{0})\times N_{A}}{2\pi }\right\rceil \leq a_{m}\leq \left\lceil \frac{(2\arctan(\frac{l_{\max}}{\sqrt{{\left(x_{k}{}^{\prime }\right)}^{2}+{\left(y_{k}{}^{\prime }\right)}^{2}}})+\theta_{0})\times N_{A}}{2\pi }\right\rceil \\ \left\lceil \frac{l_{\min}\times N_{R}}{CR}\right\rceil \leq r_{m}\leq \left\lceil \frac{l_{\max}\times N_{R}}{CR}\right\rceil \end{array}$$
where function ⌈•⌉ denotes rounding up a value and *CR* denotes the coverage range of radar. The measurements obtained from the front-end of radar are images that have *N_A_* × *N_R_* pixels, in accordance with the time series.
(7)$$Z_{k}=\{Z(r,a,t)|0\leq r\leq N_{R};1\leq a\leq N_{A};1\leq t\leq K\},$$
where *K* is the quantity of images. 

### 2.4. Problem Statement

The aim of target detection is extracting the state of targets *St_m_* by the video *Z_k_*. The quantity of video in high-resolution radar system is enormous. The parameters of the available high-resolution marine radar in this work are presented in Table 1.

Target detection must be completed in a radar scanning cycle (10 s). The images of the radar in two different scenarios are presented in Figure 3. Putting the detection performance aside, CFAR-based methods [10,11,12,13] and spatiotemporal-based methods [16,17,18] spend more than 200 s to complete the detection in the two real scenarios presented in Figure 3a,b. Meanwhile, it can be seen that the video data for the two scenarios are quite different, mainly because of the location of the two radars. Scenarios 1 and 2 correspond to Radars 1 and 2 in Figure 3c. Radar 1 is located on the hillside of a peninsula. Most of the areas near Radar 1 are sea or forest, the echo intensity of which is far less than the objects in urban areas. Meanwhile, the beams of Radar 1 are obscured by the peak in some azimuth bins. Relatively fewer clutter regions emerge for these two reasons. However, Radar 2 is located at the peak of a mountain that is facing the sea. Therefore, two urban regions around Radar 2 can be illuminated by the beams and more clutter regions emerge. The video data of the two scenarios were processed by the existing detection methods. However, the methods in [8,9,10,11,12,15,16,17] are far from meeting the real-time processing requirement. 

Meanwhile, an image of the whole surveillance area is about 200 Mb. It is impossible to directly store the past images using the methods in [16,17,18]. The methods in [16,17,18] have difficulty being performed on the current hardware (MPC8640D PowerPC). The efficient methods in our former work [19,20,21] were developed to meet the requirement of real-time processing and low memory. However, we found that the methods [19,20,21] are insufficient to cope with the complex environment. Based on those methods [16,17,18,19,20,21,22], we propose a novel detection framework, which is promising to achieve the three requirements simultaneously.

## 3. Proposed Methods

### 3.1. Sampling-Based Spatiotemporal Thresholding Method

Some clutter regions originate from some huge fixed objects such as buildings and islands. Areas with high sea conditions are also responsible for clutter regions. Clutter regions are much larger than the resolution cell. Meanwhile, as shown by the direct-viewing explanation in Figure 3a, the measurements are spatially correlated. A sampling-based spatiotemporal thresholding algorithm is proposed with the utilization of the spatial context. The implementation of the method consists of the following steps. The input of the method is *K* successive images, each of which has *N_R_* × *N_A_* pixels. The result is a sampled thresholding map. 

Step 1. The sample intervals in range and azimuth, *d_R_* and *d_A_*, are estimated according to the parameters of the radar and the size of clutter sources. The values of the two sample intervals can be set to *d_R_* and *d_A_* when the clutter region in the image is no larger than 2*d_R_* × 2*d_A_*. The sample interval in time *d_t_* is related to the variation rate of clutter regions. A larger *d_t_* can be set when the area and intensity of the clutter regions are changing slowly.

Step 2. To efficiently monitor the variation of clutter regions, only some of the pixels are uniformly selected from the images. The selected pixels used to evaluate the clutter ***Z***_m_ are called marked cells here.
(8)$$Z_{m}=\{Z(id_{R},jd_{A},nd_{t})|1\leq i\leq \frac{N_{R}}{d_{R}};1\leq j\leq \frac{N_{A}}{d_{A}};1\leq n\leq \frac{K}{d_{t}}\}$$

Step 3. The sampled spatiotemporal thresholding map *M* has (*N_R_/d_R_*) × (*N_A_/d_A_*) pixels. The intensity of the pixels can be estimated by the marked cells ***Z****_m_*. A (2*w* + 1) × (2*w* + 1) × (*K/d_t_*) local patch is defined in the marked cells. The set of marked cells in the local patch can be regarded as Equation (9) when evaluating the threshold of a marked cell (*r*, *a*):(9)$$\left\{Z(r+id_{R},a+jd_{A},K-ud_{t})|-w\leq i\leq w;-w\leq j\leq w;0\leq u\leq K/d_{t}\right\}$$

Then, the mean intensity of the local image patch at cell (*r*, *a*) is represented by *m*(*r*, *a*):(10)$$m(r,a)=\frac{1}{\frac{K-1}{d_{t}}({\left(2w+1\right)}^{2}-1)}\sum_{t=0}^{\frac{K-1}{d_{t}}}\sum_{i=-w}^{w}\sum_{j=-w}^{w}Z(r+id_{R},a+jd_{A},K-ud_{t})$$

Step 4. After obtaining the sampled spatiotemporal thresholding map *M*, the intensity of non-marked cell can be estimated by the two-dimensional linear interpolation in Equation (11).
(11)$$m(r,a)\;=\;\left[\begin{array}{l} \left(\frac{r_{2}-r}{r_{2}-r_{1}})\cdot (\frac{a_{2}-r}{a_{2}-a_{1}})\cdot m(r_{1},a_{1})+(\frac{r_{2}-r}{r_{2}-r_{1}})\cdot (\frac{a_{2}-r}{a_{2}-a_{1}})\cdot m(r_{1},a_{2}\right)\\ +(\frac{r_{2}-r}{r_{2}-r_{1}})\cdot (\frac{a_{2}-r}{a_{2}-a_{1}})\cdot m(r_{2},a_{1})+(\frac{r_{2}-r}{r_{2}-r_{1}})\cdot (\frac{a_{2}-r}{a_{2}-a_{1}})\cdot m(r_{2},a_{2}) \end{array}\right],$$
where (*r*_1_*,a*_1_), (*r*_1_*,a*_2_), (*r*_2_*,a*_1_) and (*r*_2_*,a*_2_) are the four nearest marked cells. 

The result of the sampling-based spatiotemporal thresholding method is the thresholding map *M*. Meanwhile, it is worth noting that not all of the intensities of non-marked cells are necessary for fine detection. The intensities of non-marked cells are calculated only when an extended target potentially exists in the area. Only (*N_R_/d_R_*) × (*N_A_/d_A_*) × (*K/d_t_*) cells involved in evaluating the sampled map are stored in the processer. Compared with existing spatiotemporal-based methods [16,17,18], many calculations can be saved. Meanwhile, fewer involved cells also means a drastic decrease in computation.

### 3.2. The Proposed Detection Framework

After the proposed approach above, the spatiotemporal thresholding map that has all *N_R_* × *N_A_* cells*,*
***M***
*=* {*m*(*r*,*a*)|1<*r*<*N_R_*,1<*a*<*N_A_* }, is available to detect the targets in theory. A contrast map ***C*,** which has *N_R_* × *N_A_* pixels, is defined first, and the intensity of cell (*r*, *a*) in the contrast map is denoted by *C*(*r*, *a*): (12)C(r,a)=Z(r,a,K)−m(r,a)

The input of the proposed detection framework is the contrast map *C*. Similar to the thresholding map *M*, the intensities of cells are not calculated if unnecessary. The proposed detection framework consists of two stages, coarse detection and fine detection. 

In coarse detection stage, some cells are uniformly selected from the contrast map for efficiency. The input of the coarse detection is the contrast map. 

Step 1. The sample intervals in range and azimuth, *d_r_* and *d_a_*, are estimated according to the parameters of the radar and the size of the targets. 

Step 2. The approximate locations of targets are found efficiently by uniformly selecting some of the cells from the contrast map *C*. The selected cells, *C*_s_, are called “seed cells”.
(13)$$C_{s}=\{C(id_{r},jd_{a})|1\leq i\leq \frac{N_{R}}{d_{r}};1\leq j\leq \frac{N_{A}}{d_{a}}\}$$

Step 3. The candidate areas where targets may exist are found by setting a threshold *T_d_* to the seed cells.
(14)$$\{\begin{array}{l} C(id_{r},jd_{a})\geq T;\;\mathrm{target}\;\mathrm{may}\;\mathrm{exist}\;\mathrm{in}\;(id_{r},jd_{a})\\ C(id_{r},jd_{a})<T;\;\mathrm{no}\;\mathrm{targets}\;\mathrm{exist}\;\mathrm{in}\;(id_{r},jd_{a}) \end{array}$$

The function of false alarm rate *P_FA_*(*T_d_*) and the function of target detection rate *P_D_*(*T_d_*) can be derived when the parameters of radar and targets are given. The optimal threshold can be obtained by Equation (15).
(15)$$T_{d}=\arg\max(P_{D}(T_{d})-\;P_{FA}(T_{d})),$$


The derivation and simulation of the expressions for *P_FA_*(*T_d_*) and *P_D_*(*T_d_*) can be found in our previous work [20]. 

The results of the coarse detection are the seed cells whose intensities are larger than the threshold. The set of the seed cells is assumed to have *N*_s_ elements, i.e. *C*_T_ = {*C*_s_^i^,1 ≤ *i* ≤ *N*_s_}. 

The second stage is fine detection. The accurate statement of targets is estimated by the seed set *C_T_*. The fine detection consists of the following steps. 

Step 1. A seed cell in *C*_T_ is taken to find the contour of the candidate target in contrast map *C*. The multiple contour tracking method in [19] is utilized to obtain the contours of the area under different thresholds. 

Step 2. The area can be grouped into four categories by its contours. If the area is a huge plain without outstanding peaks, the area very likely is unresolved clutter. If the area is larger than a normal target and has several outstanding peaks, the image of the area usually originates from several nearby targets. Then, go to Step 3 for further processing the area. If the area is moderate in size and has an outstanding peak, the image of the area should originate from a single target. Then, go to Step 4. If the area is very small, it is a false alarm. Then, go back to Step 1.

Step 3. The image of multiple targets is partitioned into smaller subareas, each of which can be regarded as a single target. The multilevel thresholding method using the Rain algorithm in [20] has been developed for this purpose. After obtaining the subareas, go to Step 4.

Step 4. The state of a single target can be estimated by the image of the area. The state includes not only location, size, and posture of the target, but also the texture of the subarea. The texture is promising for improving the association in multi-target tracking [27]. Then, go to Step 1 to process the next seed cell in *Cs*.

To have a better description of the proposed framework, two points are worth noting. The first is the sample intervals in spatiotemporal thresholding method and stage of coarse detection. Figure 4 presents an example of this relationship. The sample intervals *d_R_* and *d_A_* are utilized to locate the area of a clutter region. Therefore, *d_R_* and *d_A_* are larger than *dr* and *da*, which are utilized to locate the targets because the clutter regions are much larger than the targets. The sample intervals *d_R_*, *d_A_*, *d_r_*, and *d_a_* are related to the parameters of the radar and the size of targets. It assumes that the long axes of an extended target and a clutter region are *l_m_*^’^ and *L_m_*^’^. The lower limits of *l_m_*^’^ and *L_m_* are *l*_min_ and *L*_min_. Then, according to Equation (6), there are at least *d*_a_^t^ azimuth bins and at least *d*_r_^t^ range bins whose amplitudes are affected by an extended target.
(16)$$\{\begin{array}{l} d_{a}^{t}=\left\lceil \frac{(2\arctan(\frac{l_{\min}}{\sqrt{{\left(x_{k}{}^{\prime }\right)}^{2}+{\left(y_{k}{}^{\prime }\right)}^{2}}})+\theta_{0})\times N_{A}}{2\pi }\right\rceil \\ d_{r}^{t}=\left\lceil \frac{l_{\min}\times N_{R}}{CR}\right\rceil \end{array}$$

Similarly, there are at least *d*_A_^C^ azimuth bins and at least *d*_R_^C^ range bins whose amplitudes are affected by a clutter region.
(17)$$\{\begin{array}{l} d_{A}^{C}=\left\lceil \frac{(2\arctan(\frac{L_{\min}}{\sqrt{{\left(x_{k}{}^{\prime }\right)}^{2}+{\left(y_{k}{}^{\prime }\right)}^{2}}})+\theta_{0})\times N_{A}}{2\pi }\right\rceil \\ d_{R}^{C}=\left\lceil \frac{L_{\min}\times N_{R}}{CR}\right\rceil \end{array}$$

The sample intervals *d_R_*, *d_A_*, *d_r_*, and *d_a_* should be no less than the lower limits, i.e.,
(18)$$\{\begin{array}{l} d_{A}^{C}>d_{A}\\ d_{R}^{C}>d_{R} \end{array};\{\begin{array}{l} d_{a}^{t}>d_{a}\\ d_{r}^{t}>d_{r} \end{array}$$

The second point is the multiple contours of the candidate area. As presented in Figure 5, the contours of a single target, nearby targets and false alarm are represented by the black lines of different intensities. The contour of the false alarm is small and irregular. The outstanding peaks in the area of targets can be found by the contours. 

The flowchart of the proposed detection framework is presented in Figure 6. The inputted video and current image are presented in the red dashed box. The video data contain enormous cells to be processed in detection algorithms. The sampled video for spatiotemporal thresholding method is presented in the blue dashed box. Only a few cells need to be stored in the processor, thus much memory is saved. The (*N_R_/d_r_*) × (*N_A_/d_a_*) selected cells utilized in the stage of coarse detection are presented in the green dashed box. The cells involved in the fine detection are presented in the purple dashed box. The state of targets is estimated using only these cells. A small quantity of involved cells brings a significant decrease in calculations. The black dashed boxes in the bottom of Figure 6 infer that the areas are clustered into three categories and the points regarding the location of targets can be obtained. The points are the results of the target detection.

## 4. Experiment and Results

### 4.1. Real Data

Suitable memory requirement and real-time performance are two basic requirements in target detection. However, it is hard to balance good detection ability and these two requirements at the same time. To evaluate the superiority of the proposed framework in memory requirement and calculation, two problems of several representative methods are discussed in this section.

The calculation of CFAR-based methods is closely related to the quantity of the cells employed for estimating a threshold. Figure 7 shows the quantity of the cells used in several methods. The x-, y-, and z-axes in Cartesian coordinates represent azimuth, range and time axes, respectively. The blue and red cells, respectively, denote the pixels in the current frame and past frames. The green cell represents the cell whose threshold is being estimated.

Parameters *m* and *n* are the size of one target in azimuth and range. Then, the local region size is (*m* + 2*d*_1_) × (*n* + 2*d*_2_). The guard area with *m* × *n* cells exists so that the clutter pixels are collected some distance away from the test cell and target pixels are prohibited from contaminating clutter statistics estimation. The target size in the image can be estimated by the models in Section 2.3. The parameters of the radar in this work are listed in Table 1 and presented in Figure 7. *m* and *n* equal 21 and 3. respectively. Here, we set *d*_1_ = 5 and *d*_2_ = 4. *d*_1_ and *d*_2_ denote the width of protection cells on range and azimuth axes, respectively. *d*_1_ and *d*_2_ should meet the criterion in Equation (19) to ensure that the selected cells do not belong to the target.
(19)$$\{\begin{array}{l} d_{2}>d_{a}^{t}/2\\ d_{1}>d_{r}^{t}/2 \end{array}$$

However, large values of *d*_1_ and *d*_2_ mean more cells would be employed to evaluate the threshold. Then, in Cell-Averaging CFAR (CA CFAR) [28] and OS CFAR [10], (*m* + 2*d*_1_) × (*n* + 2*d*_2_) – *m* × *n* cells are employed for estimating the threshold. In CM CFAR [15], the *p* past cells at the same location are employed. We set *p* = 15 here. In spatiotemporal CFAR [16], both the (*m* + 2*d*_1_) × (*n* + 2*d*_2_) – *m* × *n* cells in current frame and the *p* cells in past frame are employed for one threshold. In spatiotemporal CA CFAR, all cells in this region are necessary, i.e. (*m* + 2*d*_1_) × (*n* + 2*d*_2_) – *m* × *n* cells in the current frame and (*m* + 2*d*_1_) × (*n* + 2*d*_2_) × *p* cells in the past frames. In the proposed framework, as presented in Figure 7, for the sampling, *a* × *b* × *c* cells are selected from the (*a* × *d_R_*) × (*b* × *d_A_*) × (*c* × *d_t_*)-sized cube. *a, b,* and *c* equal 11, 5 and 8 here, respectively. The sample intervals *d_R_*, *d_A_* and *d_t_* are 5, 10 and 5, respectively.

The quantity of employed cells for a threshold in these methods is listed in Table 2 for a better description. The sufficient quantity of employed cells and the huge distance between the test cell and target pixels guarantee that a more suitable threshold can be obtained. Meanwhile, employing more cells can improve the robustness in estimating threshold. However, it means that more calculations are spent on one threshold. In CFAR-based approaches, the threshold of all cells, *N_R_* × *N_A_* here, is calculated. However, only the threshold of (*N_R_/d_R_*) × (*N_A_/d_A_*) marked cells are estimated in our method. We set *d_R_* = 5 and *d_A_* = 10 here. Thus, only *N_R_* × *N_A_/*50 thresholds are estimated. The total quantity of employed cells equals the number of cells for one threshold and the marked cells, i.e. 440 × (*N_R_/d_R_*) × (*N_A_/d_A_*). Therefore, considerable calculations are saved. The total number of employed cells is presented in the fourth column of Table 2. 

Next, the memory requirement of the methods is compared. In OS CFAR and CA CFAR, only the current frame is utilized. The memory requirement of the two approaches is *N_R_* × *N_A_* cells. In spatiotemporal CFAR [17], CM CFAR [15] and spatiotemporal CA CFAR, the (*p–*1) past frames are also required. Therefore, the memory requirement of the three approaches is *N_R_* × *N_A_* × *p* cells. In the proposed frame, the current frame and (*N_R_/d_R_*) × (*N_A_/d_A_*) × (*c–*1) cells in past frames are necessary. The expression and the value of the memory requirement are listed in Table 3. It infers that the memory requirements of the spatiotemporal CFAR [17], CM CFAR [15] and spatiotemporal CA CFAR are much larger than the others.

After the theoretical analysis in calculation and memory requirement, we conducted an experiment in which the data for the two scenarios presented in Figure 3 were processed. The methods were performed on the PowerPC MPC8640D, 1.0 GHz with 4 GB RAM in Wind River Workbench 3.2 environment. As is presented in Table 4, it is no surprise that the elapsed time of the proposed framework was much less than the others. The CA CFAR had the second lowest calculation time for its strategy in estimating the threshold. A huge elapsed time was spent in OS CFAR [10] for sorting the cells and estimating the threshold iteratively. The elapsed time of Scenario 2 was much larger than that of Scenario 1 because Scenario 2 is more complex. More iterations were spent estimating each threshold. However, the elapsed time of the other methods was stable in different scenarios. The interval of two scans is 10 s in this radar. It is apparent that the proposed method is the only approach that can satisfy the real-time requirement.

However, real data cannot be utilized to evaluate the tracking performance, mainly because the state of the targets in the two scenarios is unknown. Even if a trajectory were obtained, we would not know that the trajectory originated from a target or a clutter resource. Therefore, synthetic data were applied in the following experiment.

### 4.2. Synthetic Data

Extensive experiments were conducted to verify the performance of the proposed framework from the robustness against to noise, the ability of background suppression and target detection ability, and the computation time of the algorithm. To fully access the superiority of the proposed algorithm, the five approaches used in Section 4.1 were included for comparison. 

In this example, a fleet constituted by three paralleled vessels is regarded as one group. The configuration of the fleet is shown in Figure 8a. The long axis and short axis of the three targets are 60 m and 15 m, respectively. The space between the two targets is 35 m. The three targets, whose initial positions are (12,225 m, 370 m), (12,175 m, 370 m) and (12125 m, 370 m), respectively, move at constant velocity (x = 5 m/s and y = 0 m/s) from t = 1 s to t = 200 s. The scanning period *T* equals 10 s for 20 scanning periods to generate the video data of the high-resolution radar with the model mentioned in Section 2 (cf. [20,21]). The echoes of three targets among 20 scans are presented in Figure 8b. The three targets can be fine-detected when the sea clutter and measurement noise are absent. Therefore, the synthetic scenario is presented in Figure 8c. In our former work [20], we should that synthetic data are similar to real data in both distribution and texture. Therefore, synthetic data are suitable to evaluate the detection performance. The seven groups of targets are moving in a surveillance area which consists of 13 subareas. The intensity of K distributed sea clutter in each subarea is different. The parameter *v* represents the shape parameter in K distributed sea clutter [26]. A larger value of *v* means a higher sea clutter. The detection rate and tracking precision can be greatly deteriorated in this situation. The synthetic images of the three targets under various shape parameter are shown in Figure 8c. The values of the shape parameter in each subarea can be seen in Figure 8c. It is worth noting that, in Group 0, no clutter exists and the only noise is the measurement errors. The study is presented to show the deterioration in performance originated from clutter regions. The targets are hard to follow by the naked eye when *v* equals 10 or 12. The synthetic video data are fed to the detection approaches for the points representing the position of a possible target. Then, as presented in Figure 1, the points are associated to form trajectories using the target-tracking approaches. In the experiment, the points obtained by the detection methods were fed to the PHD filter [1]. The result, i.e. the trajectories of targets, were obtained finally. A better set of trajectories can be achieved when an outstanding detection approach is employed. The optimal sub-pattern assignment (OSPA) distance [29] was used for evaluating the correctness of the trajectories. A lower OSPA distance means a more appropriate result. The OSPA distance between the ground truth of *n* targets ***T***
*=* {***T***_1_, ***T***_2_,…,***T**_n_*} and the estimated positions ***p***
*=* {***p***_1_, ***p***_2_,…, ***p****_n_*} in each scan can be calculated by:(20)$$\mathrm{OSPA}(T,p)=\{\begin{array}{c} D_{p,c}(T,p),m>n\\ D_{p,c}(p,T),m\leq n \end{array}$$
(21)Dp,c(T,p)=(1n(minκ∈Ω∑i=1m(dc(Ti,pκ(i)))p+(n−m)cp))1p,m≤n
where Ω represents the set of permutations of length *m* with elements taken from ***T***. The cut-off value *c* and the distance order *p* of OSPA distance were set as *c* = 100 and *p* = 1.5. Note that the cut-off parameter *c* determines the relative weighting given to the cardinality error component against the localization error component. Smaller values of *c* tend to emphasize localization errors and vice versa. 

Figure 9 shows the OSPA distance for 20 scans and also reveals that the proposed detection approach performs better than the others. Comparison between Groups 1–7 and Group 0 infers that the tracking performance would be greatly deteriorated by the clutter. The tracking performance would be further deteriorated when the intensity of clutter is high. However, the performance advantage of the proposed approach appears more obvious in severe scenarios (e.g., Group 6) because it has a strong capability of background suppression and target detection. 

The existing CFAR detectors are insufficient in the detection of the fleet for several drawbacks. Firstly, in the CA CFAR and OS CAFR detectors, the cells of Target 1 may be employed to estimate the threshold of a cell which belongs to Target 2. The cells of other targets usually have a larger intensity than the normal background cells. For the cells of another target, a higher threshold is obtained. This would decrease the detection rate of targets in the stage of target detection. A satisfactory trajectory can be hardly achieved by few points of targets, even though an advanced target tracking algorithm is utilized. Secondly, for the existence of sea clutter, two or three targets would be regarded as one large target when the intensity of the cells between two targets is high. Taking the example in Figure 8c, Targets 1 and 2 in the patch of Group 4 would be regarded as a large target for the sea clutter. Instead of two individual points, one inaccurate point is obtained. Misdetection and a huge localization error would arise in target tracking. However, in the proposed approach, for the utilization of the Rain algorithm [22], the image of multiple targets is partitioned into separate subareas where each smaller subarea denotes one potential target. Therefore, two accurate points denoting the two targets can be obtained. Thirdly, for the limitation of memory space, saving many images in the cache is impossible. Therefore, compared to the cells that can be employed in the current frame, fewer cells in past frames can be employed in CM CFAR detector and spatiotemporal-based CFAR detectors.

Meanwhile, the performance of the CFAR-based method is related to the parameters of the radar and targets. The guard cells in azimuth and range were set to match the size of the target in this experiment. Therefore, it is hard to achieve a satisfying result in real engineering because the targets have various sizes. Setting multiple guard areas in different sizes is a promising way to alleviate the problem. However, it requires many more calculations. Meanwhile, the prior information of targets is unnecessary in selecting the employed cells in the proposed approach. Therefore, the proposed approach can solve the difficulty and is superior to the others in complex environments.

The average OSPA distance of different groups is presented in Table 5. The lowest OSPA distance in each group is emphasized in boldface. It is obvious that, with the utilization of the proposed detection framework, a better target tracking result can be achieved.

The results show that the methods in [15,17] that consider several past frames are superior to the others because the clutter intensity of the cell can be estimated by the cells in the past frames, in addition to the cells in current frame. Although OS-CFAR is more appropriate in a multi-target situation, it is still possible that both the employed cells and the cell under estimation are occupied by the same extended target. A higher threshold would be obtained, which is harmful for reaching a remarkable detection rate. The problem has been relieved by considering more cells in past frames. However, there is no free lunch. The methods in [15,17] need far more memory space than OS-CFAR and CA-CFAR to store the video data of past frames. As to the proposed framework, the method is designed to detect the extended targets. Non-extended targets would be missed for the sampling. 

The average running time of the performed methods is presented in Table 6. The lowest elapsed time in each group is emphasized in boldface. The result matches the analysis in Section 4.1. The elapsed time of the proposed approach was less than the 1/80 of that of CA CFAR detector. In the proposed approach, processing one frame of synthetic image takes 0.5 ms at most. Meanwhile, the calculation of the OS CFAR detector is still much larger than those of the others. 

It can be verified by the experiment presented in this section that the proposed approach is superior to the existing detectors in regards to detection performance, calculation and memory requirement simultaneously. The outstanding detection framework is a promising approach to improve target tracking performance in real engineering. 

## 5. Conclusions

In this research, we present a target detection framework based on sampling and spatiotemporal detection. The coarse detection guarantees the real-time and low memory requirements by locating the area where targets may exist in advance. The fine detection can improve the detection performance by identifying and processing the single target, dense targets and sea clutter using different strategies. The extensive experiments showed that excellent performance, real-time processing and low memory requirement can be achieved simultaneously by the proposed detection framework. The tracking performance can be improved by utilizing the proposed approach with far fewer calculations and less memory being spent. Meanwhile, far less prior information, such as the extension of targets, is necessary in using the proposed approach. It also makes the proposed approach more practical in real engineering.

## Figures and Tables

**Figure 1 sensors-19-02912-f001:**
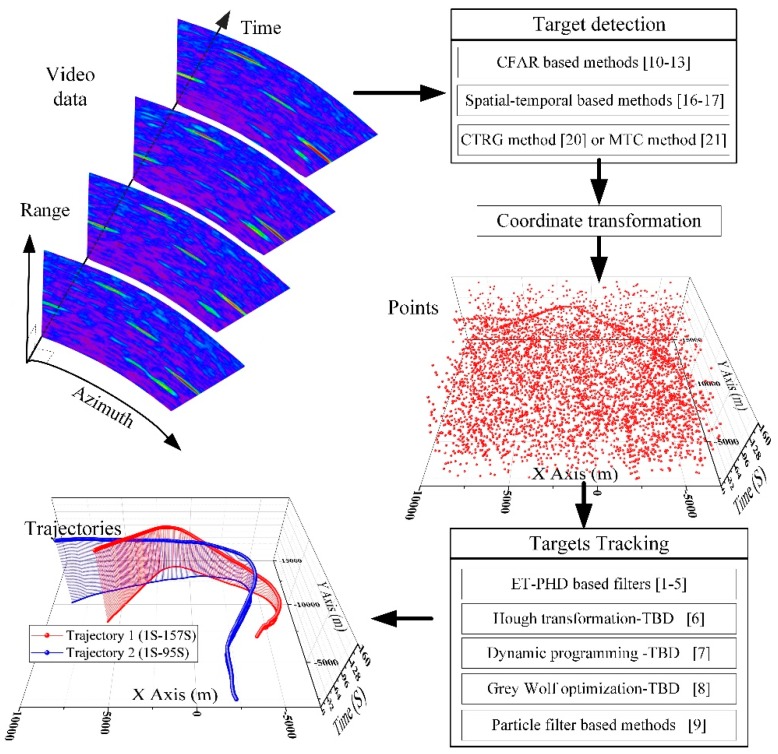
The schematic diagram of radar data processing.

**Figure 2 sensors-19-02912-f002:**
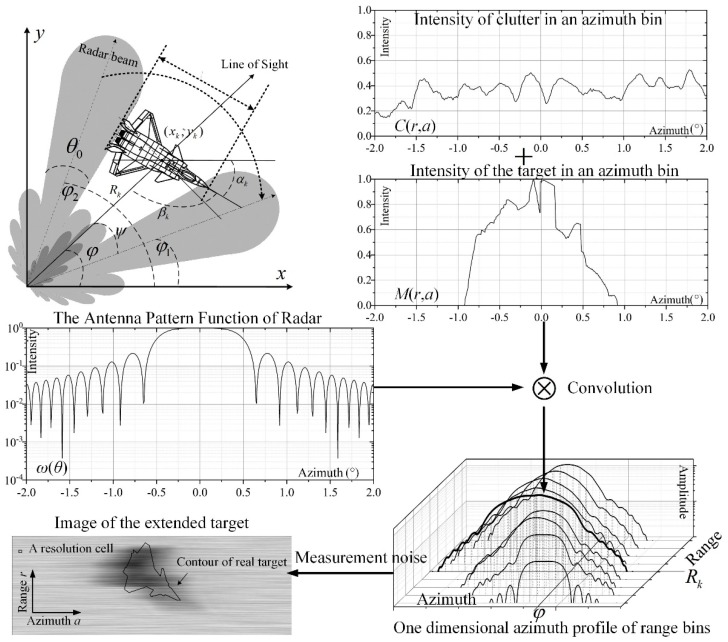
The measurement model of marine radars.

**Figure 3 sensors-19-02912-f003:**
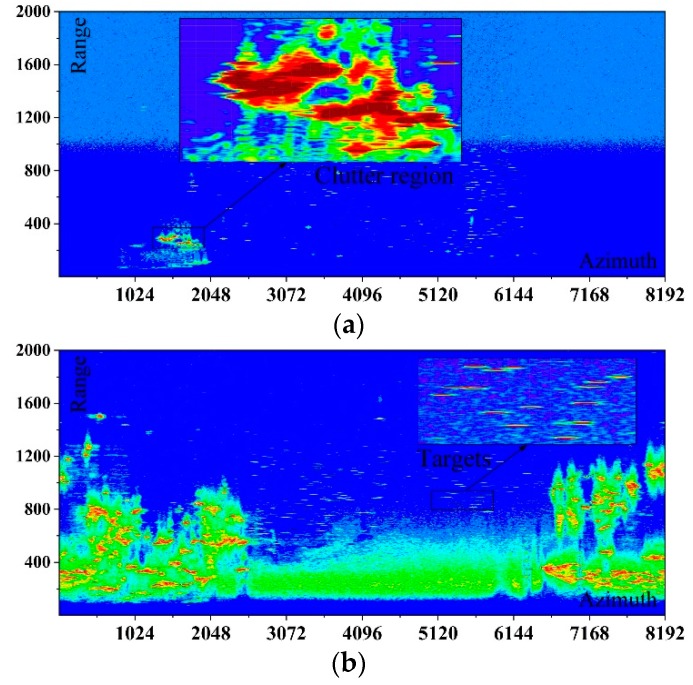
The radar image of two real scenarios: (**a**) video data of Radar 1; (**b**) video data of Radar 2; and (**c**) location of the two radars.

**Figure 4 sensors-19-02912-f004:**
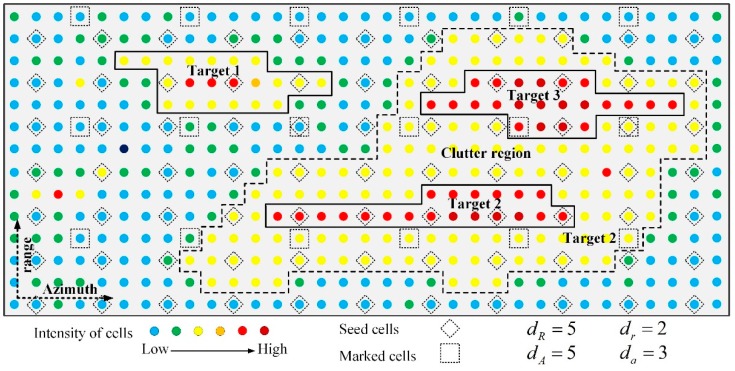
A sample of the spatiotemporal thresholding method.

**Figure 5 sensors-19-02912-f005:**
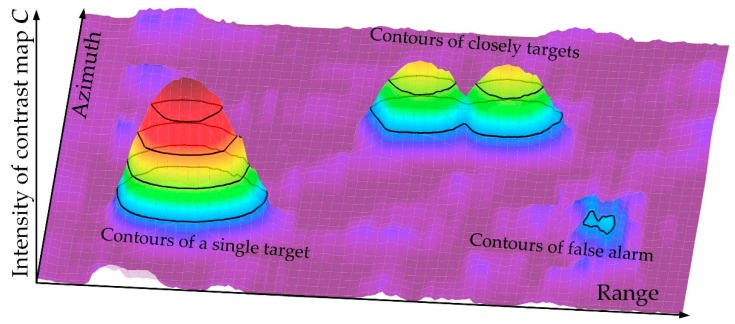
A sample of multiple contour tracking method.

**Figure 6 sensors-19-02912-f006:**
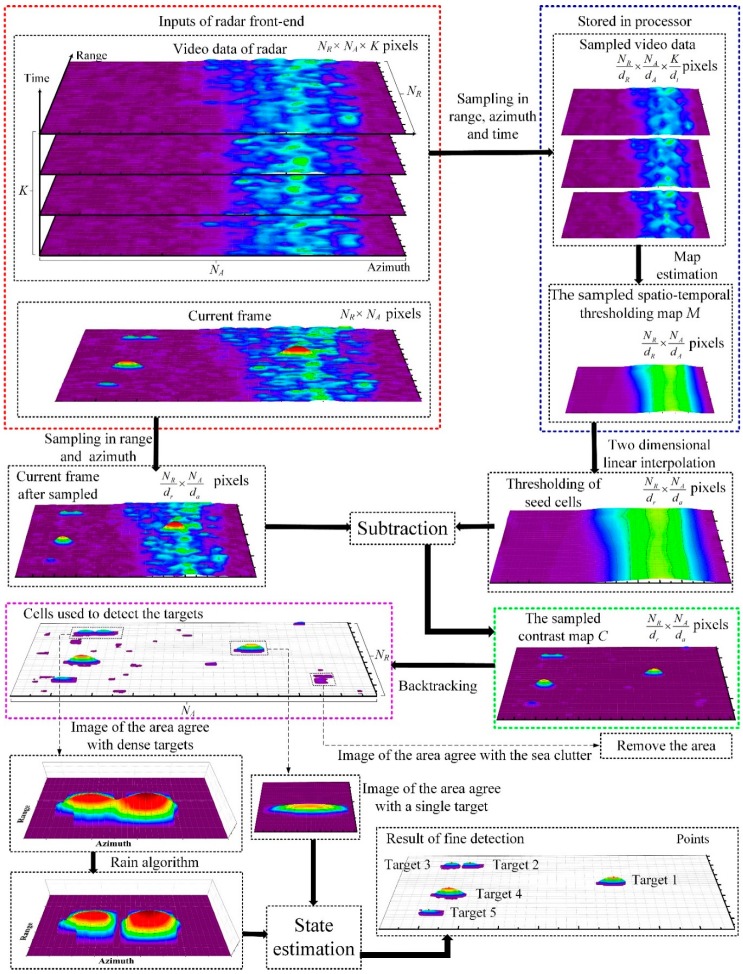
The schematic diagram of the proposed detection framework.

**Figure 7 sensors-19-02912-f007:**
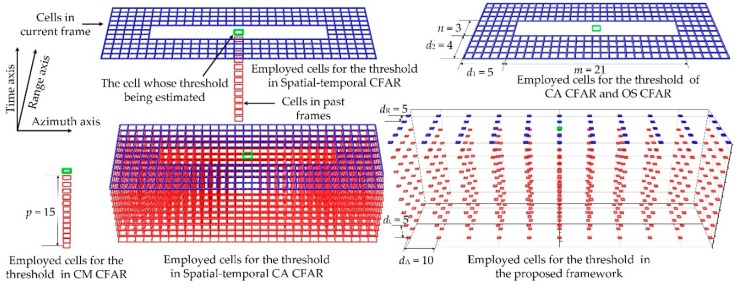
The employed cells for the detection threshold in mentioned approaches.

**Figure 8 sensors-19-02912-f008:**
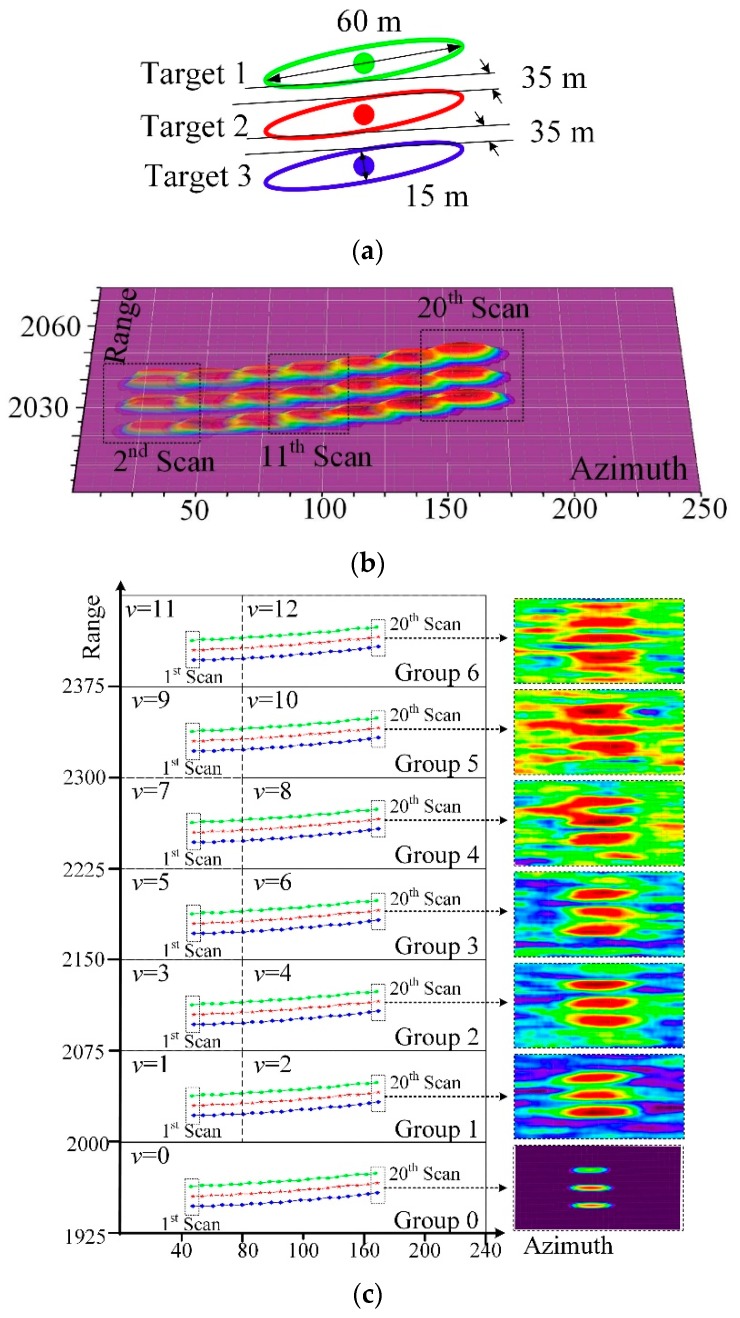
The synthetic data in this work: (**a**) the configuration of the fleet; (**b**) the echoes of the three targets among 20 scans; and (**c**) the synthetic scenario.

**Figure 9 sensors-19-02912-f009:**
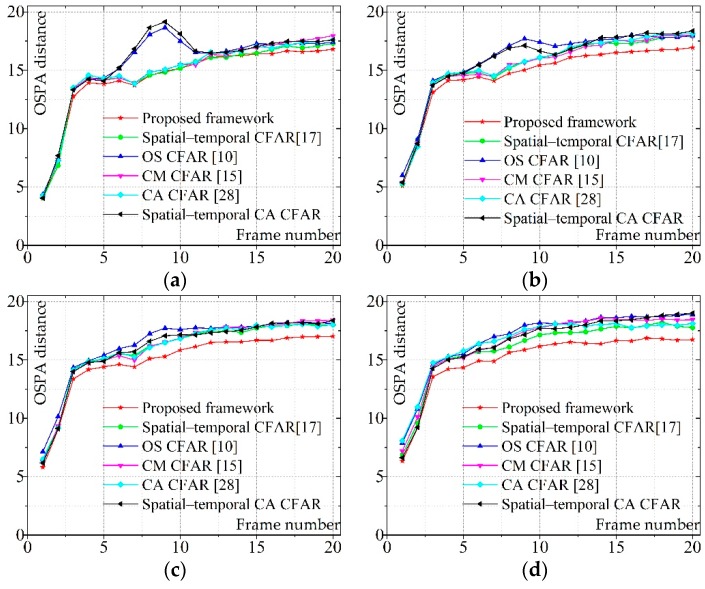
The OSPA distance of six scenarios at each scan: (**a**–**f**) Groups 1–7; and (**g**) Group 0.

**Table 1 sensors-19-02912-t001:** Parameters of the radar.

Parameter	Value
3dB azimuth beam width	0.94°
Number of bins in range axis (*N_R_*)	8192
Number of bins in range axis (*N_A_*_)_	8192
Angular Precision	0.0439°
Range Resolution	6(m)
Central frequency	1.35(GHz)
Rotating speed of antenna	π/5(°/s)

**Table 2 sensors-19-02912-t002:** The quantity of employed cells for thresholds.

	The Quantity of Cells for One Threshold	The Value of the Quantity	The Total Number of Employed Cells
The proposed framework	*a* × *b* × *c*	440	8.8 × *N_R_* × *N_A_*
Spatiotemporal CFAR [17]	(*m* + 2*d*_1_) × (*n* + 2*d*_2_)-*m* × *n + p*	293	293 × *N_R_* × *N_A_*
OS CFAR [10]	(*m* + 2*d*_1_) × (*n* + 2*d*_2_)-*m* × *n*	278	278 × *N_R_* × *N_A_*
CM CFAR [15]	*p*	15	15 × *N_R_* × *N_A_*
CA CFAR [28]	(*m* + 2*d*_1_) × (*n* + 2*d*_2_)-*m* × *n*	278	278 × *N_R_* × *N_A_*
Spatiotemporal CA CFAR	(*m* + 2*d*_1_) × (*n* + 2*d*_2_) × *p*-*m* × *n*	5052	5052 × *N_R_* × *N_A_*

**Table 3 sensors-19-02912-t003:** The memory requirement.

	In theory	In the Experiment
The proposed framework	*N_R_* × *N_A_* + (*N_R_/d_R_*) × (*N_A_/d_A_*) × (*c-*1)	1.14 *N_R_* × *N_A_*
Spatiotemporal CFAR [17]	*N_R_* × *N_A_* × *p*	15 *N_R_* × *N_A_*
OS CFAR [10]	*N_R_* × *N_A_*	*N_R_* × *N_A_*
CM CFAR [15]	*N_R_* × *N_A_* × *p*	15 *N_R_* × *N_A_*
CA CFAR [28]	*N_R_* × *N_A_*	*N_R_* × *N_A_*
Spatiotemporal CA CFAR	*N_R_* × *N_A_* × *p*	15 *N_R_* × *N_A_*

**Table 4 sensors-19-02912-t004:** Elapsed time of the methods.

	Scenario 1	Scenario 2
The proposed framework	**4.76**	**4.97**
Spatiotemporal CFAR [17]	220.01	219.19
OS CFAR [10]	3213.15	8274.43
CM CFAR [15]	256.18	260.58
CA CFAR [28]	61.77	63.26
Spatiotemporal CA CFAR	467.96	467.56

**Table 5 sensors-19-02912-t005:** OSPA distance of synthetic data.

	Group 1	Group 2	Group 3	Group 4	Group 5	Group 6	Group 0
The proposed framework	**14.4**	**14.67**	**14.96**	**15.07**	**15.43**	**15.8**	**3.97**
Spatiotemporal CFAR [17]	14.59	15.35	15.88	16	16.65	17.03	4.08
OS CFAR [10]	15.5	15.97	16.33	16.83	17.4	17.76	5.06
CM CFAR [15]	14.84	15.34	15.94	16.5	17.14	17.67	4.54
CA CFAR [28]	14.83	15.45	15.9	16.51	17.09	17.37	4.86
Spatiotemporal CA CFAR	15.6	15.81	15.97	16.4	16.9	17.47	4.84

**Table 6 sensors-19-02912-t006:** Elapsed time of the methods.

	Group 1	Group 2	Group 3	Group 4	Group 5	Group 6	Group 0
The proposed framework	**0.49**	**0.43**	**0.39**	**0.35**	**0.37**	**0.38**	**0.35**
Spatiotemporal CFAR [17]	61.36	61.68	62.06	62.11	61.84	62.39	55.6
OS CFAR [10]	5041.27	4993.38	5267.96	5272.79	5319.96	5337.25	4710.06
CM CFAR [15]	150.09	148.54	147.95	149.52	148.38	147.67	133.54
CA CFAR [28]	37.64	37.68	38.11	38.08	38.04	38.08	34.12
Spatiotemporal CA CFAR	400.43	397.42	402.07	400.77	383.4	384	355.58

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
