# Peer review of "An Efficient Extended Targets Detection Framework Based on Sampling and Spatio-Temporal Detection"

_sensors, 2019, doi:10.3390/s19132912_

Round 1
Reviewer 1 Report
The paper presents a target detection framework based on sampling and spatio-temporal detection with real-time processing and low memory requirement. The presented experiments sustain the performance of the proposed method.
The experiments are made using both real data and synthetic data.
For the real data - images from the radar for two different scenarios are used. Please explain the scenarios. What are the cases that are relevant for these scenarios? Can be extended the obtained results to other scenarios that are different from the two used in the evaluation. Please explain how was chosen values d1 = 5, d2 = 4.
Please explain why synthetic data are used for the evaluation. Describe that cases that are covered by the synthetic data. Why are synthetic data used instead of real data?
Author Response
Thank you very much for your comments about our paper submitted to Sensors (sensors-497093),”an efficient extended targets detection framework based on sampling and spatio-temporal detection”. We have checked the manuscript and revised it according to the comments. And a list of changes is patched in the docx file.

Reviewer 2 Report
The problem adressed in this manuscript can be of interest to the radar community. However, some points should be clarified before the paper is accepted for publication.
1) As it is well known, the shape parameter of the K distribution (sea clutter model) gives an indication about the state of the sea. The authors should provide the value of this parameter and add a study about the performance of their method in the case of a spiky clutter (shape parameter close to zero).
2) The use of the CA-CFAR or the OS-CFAR depends on the environment in which the radar operates. For the OS-CFAR detector, it is more appropriate in the case of a multi-target situation.
It is not clear from the manuscript in which situation the CA-CFAR and the OS-CFAR are used
3) The authors should justify the choice of the employed cells for the threshold and how the threshold is computed.
Author Response

(The authors gave the same response as above.)

Reviewer 3 Report
The paper describes a novel two-stage target detection framework, selection of seed states with subsequent selection of the appropriate candidates.
The method behind the paper is novel and reasonable, however there are some minor problems to fix:
- in section 2.4, which is called the problem statement, there is a description of used hardware. The reviewer thinks that it belongs to the experimental section, where indeed, in section 4.1, the configuration is stated again. Therefore, the proposal is to remove the repetitive statement from section 2.4
- although the article is well written, there are some minor problems with word usage:
-- "The excellent performance, real-time processing and low memory requirement can be achieved by the two stages, coarse detection and fine detection.
" and "Excellent performance can be achieved by the fine detection." -> "The performance is improved by fine detection" (as excellence is impossible to quantify)
-- Section 2.4: "Operating system program takes 2GB RAM and merely about 1.5GB can be utilized to perform the algorithm." The software uses 2GB RAM; besides of that, the reviewer thinks that this statement should belong to the experimental section, not the problem statement
-- "Part of clutter regions is originated from some fixed huge objects of no interest such as building and islands." -> buildings
-- 'Meanwhile, fewer involved cells also mean a drastic decreasing calculation.' -> drastic decrease in computation
-- Page 8: 'The data in this part are enormous.' This statement should be clarified or reworded.
-- Page 11: 'It seems that only the proposed approach can satisfy the real-time requirement.'apparently, the proposed approach amongst the ones which have been compared
Author Response

(The authors gave the same response as above.)

Reviewer 4 Report
The present method seems no more than ad-hoc thresholding/gating approaches and a combination of the authors' previous work. No solid theoretical analysis or proof has been given. It has actually been unclear what has really been done? Where does the gain come from? What is the key difference to the previous work?
Furthermore, for extended targets, it is practially insufficient (over-simplified) to model the problem only in 2-D position space only. As the relative position and direction of the targets to the sensor is changing over time, the coordinates have to take into account the 3-dimensional position.
Many language problems, e.g., "roughly areas"
Author Response

(The authors gave the same response as above.)

Round 2
Reviewer 2 Report
I think that the authors have responded to most of the concerns raised for this manuscript. I am in favour of its publication.The authors should review deeply the English.
Author Response
Thank you for your comments The English language and grammar in this manuscript have been further polished by another researcher who is skillful in scientific English. Meanwhile, the novelty of this work is further emphasized in the abstract.
Reviewer 4 Report
While I still believe that the manuscript should further highlight the novelty of the work as compared with the previous work, the authors have revised the paper according to the comments of all reviewers and prepared a fair response letter. I am convinced for the publication of the work now. The following review regarding the nonlinear particle filter for target tracking may be added to the reference list.
X. Wang, et al. A survey of recent advances in particle filters and remaining challenges for multi-target tracking, Sensors, 17(12). pii: E2707, 2017
Author Response
Thank you for your comments. We have checked the manuscript and revised it according to the comments. And a list of changes is patched below. If you have any question about this paper, please let me know. The changes in this manuscript has been highlighted with colored text. Please see the attachment.
